# Learning Distributed Representations for Structured Output Prediction

**Vivek Srikumar**[*]
University of Utah
svivek@cs.utah.edu

**Christopher D. Manning**
Stanford University
manning@cs.stanford.edu

## Abstract

In recent years, distributed representations of inputs have led to performance gains in many applications by allowing statistical information to be shared across inputs. However, the predicted outputs (labels, and more generally structures) are still treated as discrete objects even though outputs are often not discrete units of meaning. In this paper, we present a new formulation for structured prediction where we represent individual labels in a structure as dense vectors and allow semantically similar labels to share parameters. We extend this representation to larger structures by defining compositionality using tensor products to give a natural generalization of standard structured prediction approaches. We define a learning objective for jointly learning the model parameters and the label vectors and propose an alternating minimization algorithm for learning. We show that our formulation outperforms structural SVM baselines in two tasks: multiclass document classification and part-of-speech tagging.

## 1 Introduction

In recent years, many computer vision and natural language processing (NLP) tasks have benefited from the use of dense representations of inputs by allowing superficially different inputs to be related to one another [26, 9, 7, 4]. For example, even though words are not discrete units of meaning, traditional NLP models use indicator features for words. This forces learning algorithms to learn separate parameters for orthographically distinct but conceptually similar words. In contrast, dense vector representations allow sharing of statistical signal across words, leading to better generalization.

Many NLP and vision problems are structured prediction problems. The output may be an atomic label (tasks like document classification) or a composition of atomic labels to form combinatorial objects like sequences (e.g. part-of-speech tagging), labeled trees (e.g. parsing) or more complex graphs (e.g. image segmentation). Despite both the successes of distributed representations for inputs *and* the clear similarities over the output space, it is still usual to handle outputs as discrete objects. But are structures, and the labels that constitute them, really discrete units of meaning?

Consider, for example, the popular 20 Newsgroups dataset [13] which presents the multiclass classification problem of identifying a newsgroup label given the text of a posting. Labels include `comp.os.mswindows.misc`, `sci.electronics`, `comp.sys.mac.hardware`, `rec.autos` and `rec.motorcycles`. The usual strategy is to train a classifier that uses separate weights for each label. However, the labels themselves have meaning that is independent of the training data. From the label, we can see that `comp.os.mswindows.misc`, `sci.electronics` and `comp.sys.mac.hardware` are semantically closer to each other than the other two. A similar argument can be made for not just atomic labels but their compositions too. For example, a part-of-speech tagging system trained as a sequence model might have to learn separate parameters

---

[*]This work was done when the author was at Stanford University.

for the JJ→NNS and JJR→NN transitions even though both encode a transition from an adjective to a noun. Here, the similarity of the transitions can be inferred from the similarity of its components.

In this paper, we propose a new formulation for structured output learning called DISTRO (DIStributed STRuctred Output), which accounts for the fact that labels are not atomic units of meaning. We model label meaning by representing individual labels as real valued vectors. Doing so allows us to capture similarities between labels. To allow for arbitrary structures, we define compositionality of labels as tensor products of the label vectors corresponding to its sub-structures. We show that doing so gives us a natural extension of standard structured output learning approaches, which can be seen as special cases with one-hot label vectors.

We define a learning objective that seeks to jointly learn the model parameters along with the label representations and propose an alternating algorithm for minimizing the objective for structured hinge loss. We evaluate our approach on two tasks which have semantically rich labels: multiclass classification on the newsgroup data and part-of-speech tagging for English and Basque. In all cases, we show that DISTRO outperforms the structural SVM baselines.

## 1.1 Related Work

This paper considers the problem of using distributed representations for arbitrary structures and is related to recent work in deep learning and structured learning. Recent unsupervised representation learning research has focused on the problem of embedding inputs in vector spaces [26, 9, 16, 7]. There has been some work [22] on modeling semantic compositionality in NLP, but the models do not easily generalize to arbitrary structures. In particular, it is not easy to extend these approaches to use advances in knowledge-driven learning and inference that standard structured learning and prediction algorithms enable.

Standard learning approaches for structured output allow for modeling arbitrarily complex structures (subject to inference difficulties) and structural SVMs [25] or conditional random fields [12] are commonly used. However, the output itself is treated as a discrete object and similarities between outputs are not modeled. For multiclass classification, the idea of classifying to a label set that follow a known hierarchy has been explored [6], but such a taxonomy is not always available.

The idea of distributed representations for outputs has been discussed in the connectionist literature since the eighties [11, 21, 20]. In recent years, we have seen several lines of research that address the problem in the context of multiclass classification by framing feature learning as matrix factorization or sparse encoding [23, 1, 3]. As in this paper, the goal has often explicitly been to discover shared characteristics between the classes [2]. Indeed, the inference formulation we propose is very similar to inference in these lines of work. Also related is recent research in the NLP community that explores the use of tensor decompositions for higher order feature combinations [14]. The primary novelty in this paper is that in addition to representing atomic labels in a distributed manner, we model their compositions in a natural fashion to generalize standard structured prediction.

## 2 Preliminaries and Notation

In this section, we give a very brief overview of structured prediction with the goal of introducing notation and terminology for the next sections. We represent inputs to the structured prediction problem (such as, sentences, documents or images) by $\mathbf{x} \in \mathcal{X}$ and output structures (such as labels or trees) by $\mathbf{y} \in \mathcal{Y}$. We define the feature function $\Phi : \mathcal{X} \times \mathcal{Y} \to \Re^n$ that captures the relationship between the input $\mathbf{x}$ and the structure $\mathbf{y}$ as an $n$ dimensional vector. A linear model scores the structure $\mathbf{y}$ with a weight vector $\mathbf{w} \in \Re^n$ as $\mathbf{w}^T \Phi(\mathbf{x}, \mathbf{y})$. We predict the output for an input $\mathbf{x}$ as $\arg\max_{\mathbf{y}} \mathbf{w}^T \Phi(\mathbf{x}, \mathbf{y})$. This problem of *inference* is a combinatorial optimization problem.

We will use the structures in Figure 1 as running examples. In the case of multiclass classification, the output $\mathbf{y}$ is one of a finite set of labels (Figure 1, left). For more complex structures, the feature vector is decomposed over the parts of the structure. For example, the usual representation of a first-order linear sequence model (Figure 1, middle) decomposes the sequence into emissions and transitions and the features decompose over these [8]. In this case, each emission is associated with one label and a transition is associated with an ordered pair of labels.

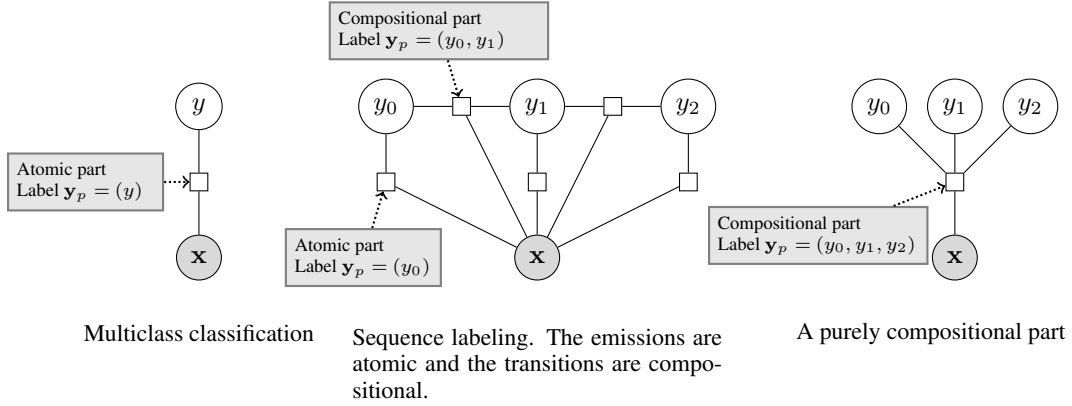

Multiclass classification

Sequence labeling. The emissions are atomic and the transitions are compositional.

A purely compositional part

Figure 1: Three examples of structures. In all cases, $\mathbf{x}$ represents the input and the $y$'s denote the outputs to be predicted. Here, each square represents a *part* as defined in the text and circles represent random variables for inputs and outputs (as in factor graphs). The left figure shows multiclass classification, which has an atomic part associated with exactly one label. The middle figure shows a first-order sequence labeling task that has both atomic parts (emissions) and compositional ones (transitions). The right figure shows a purely compositional part where all outputs interact. The *feature functions* for these structures are shown at the end of Section 3.1.

In the general case, we denote the parts (or equivalently, factors in a factor graph) in the structure for input $\mathbf{x}$ by $\Gamma_{\mathbf{x}}$. Each part $p \in \Gamma_{\mathbf{x}}$ is associated with a list of discrete labels, denoted by $\mathbf{y}_p = (y_p^0, y_p^1, \cdots)$. Note that the size of the list $\mathbf{y}_p$ is a modeling choice; for example, transition parts in the first-order Markov model correspond to two consecutive labels, as shown in Figure 1.

We denote the set of labels in the problem as $\mathcal{L} = \{l_1, l_2, \cdots, l_M\}$ (e.g. the set of part-of-speech tags). All the elements of the part labels $\mathbf{y}_p$ are members of this set. For notational convenience, we denote the first element of the list $\mathbf{y}_p$ by $y_p$ (without boldface) and the rest by $\mathbf{y}_p^{1:}$. In the rest of the paper, we will refer to a part associated with a single label as *atomic* and all other parts where $\mathbf{y}_p$ has more than one element as *compositional*. In Figure 1, we see examples of a purely atomic structure (multiclass classification), a purely compositional structure (right) and a structure that is a mix of the two (first order sequence, middle).

The decomposition of the structure decomposes the feature function over the parts as

$$\Phi(\mathbf{x}, \mathbf{y}) = \sum_{p \in \Gamma_{\mathbf{x}}} \Phi_p(\mathbf{x}, \mathbf{y}_p). \tag{1}$$

The scoring function $\mathbf{w}^T \phi(\mathbf{x}, \mathbf{y})$ also decomposes along this sum. Standard definitions of structured prediction models leave the definition of the part-specific feature function $\Phi_p$ to be problem dependent. We will focus on this aspect in Section 3 to define our model.

With definitions of a scoring function and inference, we can state the learning objective. Given a collection of $N$ training examples of the form $(\mathbf{x}^i, \mathbf{y}^i)$, training is the following regularized risk minimization problem:

$$\min_{\mathbf{w} \in \Re^n} \frac{\lambda}{2} \mathbf{w}^T \mathbf{w} + \frac{1}{N} \sum_i L(\mathbf{x}^i, \mathbf{y}^i; \mathbf{w}). \tag{2}$$

Here, $L$ represents a loss function such as the hinge loss (for structural SVMs) or the log loss (for conditional random fields) and penalizes model errors. The hyper-parameter $\lambda$ trades off between generalization and accuracy.

## 3 Distributed Representations for Structured Output

As mentioned in Section 2, the choice of the feature function $\Phi_p$ for a part $p$ is left to be problem specific. The objective is to capture the correlations between the relevant attributes of the input $\mathbf{x}$ and the output labels $\mathbf{y}_p$. Typically, this is done by conjoining the labels $\mathbf{y}_p$ with a user-defined feature vector $\phi_p(\mathbf{x})$ that is dependent only on the input.

When applied to atomic parts (e.g. multiclass classification), conjoining the label with the input features effectively allocates a different portion of the weight vector for each label. For compositional parts (e.g. transitions in sequence models), this ensures that each combination of labels is associated with a different portion of the weight vector. The implicit assumption in this design is that labels and label combinations are distinct units of meaning and hence do not share any parameters across them. In this paper, we posit that in most naturally occurring problems and their associated labels, this assumption is not true. In fact, labels often encode rich semantic information with varying degrees of similarities to each other. Because structures are composed of atomic labels, the same applies to structures too.

From Section 2, we see that for the purpose of inference, structures are completely defined by their feature vectors, which are decomposed along the atomic and compositional parts that form the structure. Thus, our goal is to develop a feature representation for labeled parts that exploits label similarity. More explicitly, our desiderata are:

1. First, we need to be able to represent labeled atomic parts using a feature representation that accounts for relatedness of labels in such a way that statistical strength (i.e. weights) can be shared across different labels.
2. Second, we need an operator that can construct compositional parts to build larger structures so that the above property can be extended to arbitrary structured output.

### 3.1 The DISTRO model

In order to assign a notion of relatedness between labels, we associate a $d$ dimensional unit vector $\mathbf{a}_l$ to each label $l \in \mathcal{L}$. We will refer to the $d \times M$ matrix comprising of all the $M$ label vectors as $\mathbf{A}$, the label matrix.

We can define the feature vectors for parts, and thus entire structures, using these label vectors. To do so, we define the notion of a *feature tensor function* for a part $p$ that has been labeled with a list of $m$ labels $\mathbf{y}_p$. The feature tensor function is a function $\Psi_p$ that maps the input $\mathbf{x}$ and the label list $\mathbf{y}_p$ associated with the part to a tensor of order $m + 1$. The tensor captures the relationships between the input and all the $m$ labels associated with it. We recursively define the feature tensor function using the label vectors as:

$$\Psi_p\left(\mathbf{x}, \mathbf{y}_p, \mathbf{A}\right) = \begin{cases} \mathbf{a}_{l_{y_p}} \otimes \phi_p(\mathbf{x}), & p \text{ is atomic,} \\ \mathbf{a}_{l_{y_p}} \otimes \Psi_p\left(\mathbf{x}, \mathbf{y}_p^{1:}, \mathbf{A}\right), & p \text{ is compositional.} \end{cases} \tag{3}$$

Here, the symbol $\otimes$ denotes the tensor product operation. Unrolling the recursion in this definition shows that the feature tensor function for a part is the tensor product of the vectors for all the labels associated with that part and the feature vector associated with the input for the part. For an input $\mathbf{x}$ and a structure $\mathbf{y}$, we use the feature tensor function to define its feature representation as

$$\Phi_{\mathbf{A}}\left(\mathbf{x}, \mathbf{y}\right) = \sum_{p \in \Gamma_{\mathbf{x}}} vec\left(\Psi_p\left(\mathbf{x}, \mathbf{y}_p, \mathbf{A}\right)\right) \tag{4}$$

Here, $vec(\cdot)$ denotes the vectorization operator that converts a tensor into a vector by stacking its elements. Figure 2 shows an example of the process of building the feature vector for a part that is labeled with two labels. With this definition of the feature vector, we can use the standard approach to score structures using a weight vector as $\mathbf{w}^T \Phi_{\mathbf{A}}\left(\mathbf{x}, \mathbf{y}\right)$.

In our running examples from Figure 1, we have the following definitions of feature functions for each of the cases:

1. Purely atomic part, multiclass classification (left): Denote the feature vector associated with $\mathbf{x}$ as $\phi$. For an atomic part, the definition of the feature tensor function in Equation (3) effectively produces a $d \times |\phi|$ matrix $\mathbf{a}_{l_y} \phi^T$. Thus the feature vector for the structure $\mathbf{y}$ is $\Phi_{\mathbf{A}}\left(\mathbf{x}, \mathbf{y}\right) = vec\left(\mathbf{a}_{l_y} \phi^T\right)$. For this case, the score for an input $\mathbf{x}$ being assigned a label $\mathbf{y}$ can be explicitly be written as the following summation:

$$\mathbf{w}^T \Phi_{\mathbf{A}}\left(\mathbf{x}, \mathbf{y}\right) = \sum_{i=0}^{d} \sum_{j=0}^{|\phi|} w_{dj+i} a_{l_y,i} \phi_j$$

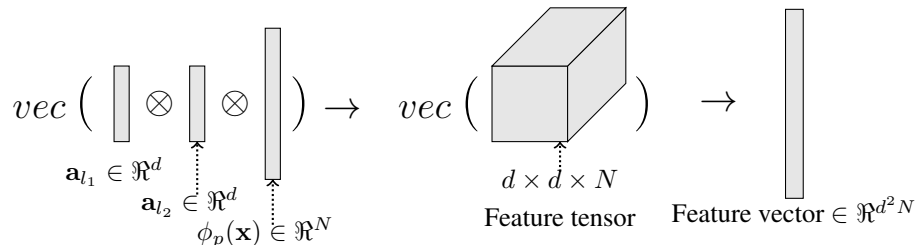

Figure 2: This figure summarizes feature vector generation for a compositional part labeled with two labels $l_1$ and $l_2$. Each label is associated with a $d$ dimensional label vector and the feature vector for the input is $N$ dimensional. Vectorizing the feature tensor produces a final feature vector that is a $d^2 N$-dimensional vector.

2. Purely compositional part (right): For a compositional part, the feature tensor function produces a tensor whose elements effectively enumerate every possible combination of elements of input vector $\phi_p(\mathbf{x})$ and the associated label vectors. So, the feature vector for the structure is $\Phi_{\mathbf{A}}(\mathbf{x}, \mathbf{y}) = vec\left(\mathbf{a}_{l_{y_0}} \otimes \mathbf{a}_{l_{y_1}} \otimes \mathbf{a}_{l_{y_2}} \otimes \phi_p(\mathbf{x})\right)$.

3. First order sequence (middle): This structure presents a combination of atomic and compositional parts. Suppose we denote the input emission features by $\phi^{E,i}$ for the $i^{th}$ label and the input features corresponding to the transition[1] from $y_i$ to $y_{i+1}$ by $\phi^{T,i}$. With this notation, we can define the feature vector for the structure as

$$\Phi_{\mathbf{A}}(\mathbf{x}, \mathbf{y}) = \sum_i vec\left(\mathbf{a}_{l_{y_i}} \otimes \phi^{E,i}\right) + \sum_i vec\left(\mathbf{a}_{l_{y_i}} \otimes \mathbf{a}_{l_{y_{i+1}}} \otimes \phi^{T,i}\right).$$

## 3.2 Discussion

**Connection to standard structured prediction**   For a part $p$, a traditional structured model conjoins all its associated labels to the input feature vector to get the feature vector for that assignment of the labels. According to the definition of Equation (3), we propose that these label conjunctions should be replaced with a tensor product, which generalizes the standard method. Indeed, if the labels are represented via one-hot vectors, then we would recover standard structured prediction where each label (or group of labels) is associated with a separate section of the weight vector. For example, for multiclass classification, if each label is associated with a separate one-hot vector, then the feature tensor for a given label will be a matrix where exactly one column is the input feature vector $\phi_p(\mathbf{x})$ and all other entries are zero. This argument also extends to compositional parts.

**Dimensionality of label vectors**   If labels are represented by one-hot vectors, the dimensionality of the label vectors will be $M$, the number of labels in the problem. However, in DISTRO, in addition to letting the label vectors be any unit vector, we can also allow them to exist in a lower dimensional space. This presents us with a decision with regard to the dimensionality $d$.

The choice of $d$ is important for two reasons. First, it determines the number of parameters in the model. If a part is associated with $m$ labels, recall that the feature tensor function produces a $m+1$ order tensor formed by taking the tensor product of the $m$ label vectors and the input features. That is, the feature vector for the part is a $d^m |\phi_p(\mathbf{x})|$ dimensional vector. (See 2 for an illustration.) Smaller $d$ thus leads to smaller weight vectors. Second, if the dimensionality of the label vectors is lower, it encourages more weights to be shared across labels. Indeed, for purely atomic and compositional parts if the labels are represented by $M$ dimensional vectors, we can show that for any weight vector that scores these labels via the feature representation defined in Equation (4), there is another weight vector that assigns the same scores using one-hot weight vectors.

## 4   Learning Weights and Label Vectors

In this section, we will address the problem of learning the weight vectors $\mathbf{w}$ and the label vectors $\mathbf{A}$ from data. We are given a training set with $N$ examples of the form $(\mathbf{x}^i, \mathbf{y}^i)$. The goal of learning

is to minimize regularized risk over the training set. This leads to a training objective similar to that of structural SVMs or conditional random fields (Equation (2)). However, there are two key differences. First, the feature vectors for structures are not fixed as in structural SVMs or CRFs but are functions of the label vectors. Second, the minimization is over not just the weight vectors, but also over the label vectors that require regularization.

In order to encourage the labels to share weights, we propose to impose a rank penalty over the label matrix $\mathbf{A}$ in the learning objective. Since the rank minimization problem is known to be computationally intractable in general [27], we use the well known nuclear norm surrogate to replace the rank [10]. This gives us the learning objective defined as $f$ below:

$$f(\mathbf{w}, \mathbf{A}) = \frac{\lambda_1}{2} \mathbf{w}^T \mathbf{w} + \lambda_2 ||\mathbf{A}||_* + \frac{1}{N} \sum_i L(\mathbf{x}^i, \mathbf{y}^i; \mathbf{w}, \mathbf{A}) \tag{5}$$

Here, the $||\mathbf{A}||_*$ is the nuclear norm of $\mathbf{A}$, defined as the sum of the singular values of the matrix. Compared to the objective in Equation (2), the loss function $L$ is also dependent of the label matrix via the new definition of the features. In this paper, we instantiate the loss using the structured hinge loss [25]. That is, we define $L$ to be

$$L(\mathbf{x}^i, \mathbf{y}^i; \mathbf{w}, \mathbf{A}) = \max_{\mathbf{y}} \left( \mathbf{w}^T \Phi_{\mathbf{A}}(\mathbf{x}^i, \mathbf{y}) + \Delta(\mathbf{y}, \mathbf{y}^i) - \mathbf{w}^T \Phi_{\mathbf{A}}(\mathbf{x}^i, \mathbf{y}^i) \right) \tag{6}$$

Here, $\Delta$ is the Hamming distance. This defines the DISTRO extension of the structural SVM.

The goal of learning is to minimize the objective function $f$ in terms of both its parameters $\mathbf{w}$ and $\mathbf{A}$, where each column of $\mathbf{A}$ is restricted to be a unit vector by definition. However, the objective is not longer jointly convex in both $\mathbf{w}$ and $\mathbf{A}$ because of the product terms in the definition of the feature tensor.

We use an alternating minimization algorithm for solving the optimization problem (Algorithm 1). If the label matrix $\mathbf{A}$ is fixed, then so are the feature representations of structures (from Equation (4)). Thus, for a fixed $\mathbf{A}$ (lines 2 and 5), the problem of minimizing $f(\mathbf{w}, \mathbf{A})$ with respect to only $\mathbf{w}$ is identical to the learning problem of structural SVMs. Since gradient computation and inference do not change from the usual setting, we can solve this minimization over $\mathbf{w}$ using stochastic sub-gradient descent (SGD). For fixed weight vectors (line 4), we implemented stochastic sub-gradient descent using the proximal gradient method [18] for solving for $\mathbf{A}$. The supplementary material gives further details about the steps of the algorithm.

---

**Algorithm 1** Learning algorithm by alternating minimization. The goal is to solve $\min_{\mathbf{w}, \mathbf{A}} f(\mathbf{w}, \mathbf{A})$. The input to the problem is a training set of examples consisting of pairs of labeled inputs $(\mathbf{x}^i, \mathbf{y}^i)$ and $T$, the number of iterations.

---
1: Initialize $\mathbf{A}^0$ randomly
2: Initialize $\mathbf{w}^0 = \min_{\mathbf{w}} f(\mathbf{w}, \mathbf{A}^0)$
3: **for** $t = 1, \cdots, T$ **do**
4:    $\mathbf{A}^t \leftarrow \min_{\mathbf{A}} f(\mathbf{w}^{t-1}, \mathbf{A})$
5:    $\mathbf{w}^{t+1} \leftarrow \min_{\mathbf{w}} f(\mathbf{w}, \mathbf{A}^t)$
6: **end for**
7: **return** $(\mathbf{w}^{T+1}, \mathbf{A}^T)$

---

Even though the objective function is not jointly convex in $\mathbf{w}$ and $\mathbf{A}$, in our experiments (Section 5), we found that in all but one trial, the non-convexity of the objective did not affect performance. Because the feature functions are multilinear in $\mathbf{w}$ and $\mathbf{A}$, multiple equivalent solutions can exist (from the perspective of the score assigned to structures) and the eventual point of convergence is dependent on the initialization.

For regularizing the label matrix, we also experimented with the Frobenius norm and found that not only does the nuclear norm have an intuitive explanation (rank minimization) but also performed better. Furthermore, the proximal method itself does not add significantly to the training time because the label matrix is small. In practice, training time is affected by the density of the label vectors and sparser vectors correspond to faster training because the sparsity can be used to speed up dot product computation. Prediction is as fast as inference in standard models, however, because the only change is in feature computation via the vectorization operator, which can be performed efficiently.

# 5 Experiments

We demonstrate the effectiveness of DISTRO on two tasks – document classification (purely atomic structures) and part-of-speech (POS) tagging (both atomic and compositional structures). In both cases, we compare to structural SVMs – i.e. the case of one-hot label vectors – as the baseline.

We selected the hyper-parameters for all experiments by cross validation. We ran the alternating algorithm for 5 epochs for all cases with 5 epochs of SGD for both the weight and label vectors. We allowed the baseline to run for 25 epochs over the data. For the proposed method, we ran all the experiments five times with different random initializations for the label vectors and report the average accuracy. Even though the objective is not convex, we found that the learning algorithm converged quickly in almost all trials. When it did not, the objective value on the training set at the end of each alternating SGD step in the algorithm was a good indicator for ill-behaved initializations. This allowed us to discard bad initializations during training.

## 5.1 Atomic structures: Multiclass Classification

Our first application is the problem of document classification with the 20 Newsgroups Dataset [13]. This dataset is collection of about 20,000 newsgroup posts partitioned roughly evenly among 20 newsgroups. The task is to predict the newsgroup label given the post. As observed in Section 1, some newsgroups are more closely related to each other than others.

We used the 'bydate' version of the data with tokens as features. Table 1 reports the performance of the baseline and variants of DISTRO for newsgroup classification. The top part of the table compares the baseline to our method and we see that modeling the label semantics gives us a 2.6% increase in accuracy. In a second experiment (Table 1, bottom), we studied the effect of explicitly reducing the label vector dimensionality. We see that even with 15 dimensional vectors, we can outperform the baseline and the performance of the baseline is almost matched with 10 dimensional vectors. Recall that the size of the weight vector increases with increasing label vector dimensionality (see Figure 2). This motivates a preference for smaller label vectors.

| Algorithm | Label Matrix Rank | Average accuracy (%) |
|---|---|---|
| Structured SVM | 20 | 81.4 |
| DISTRO | 19 | **84.0** |
| Reduced dimensionality setting | | |
| DISTRO | 15 | 83.1 |
| DISTRO | 10 | 80.9 |

Table 1: **Results on 20 newsgroup classification.** The top part of the table compares the baseline against the full DISTRO model. The bottom part shows the performance of two versions of DISTRO where the dimensionality of the label vectors is fixed. Even with 10-dimensional vectors, we can almost match the baseline.

## 5.2 Compositional Structures: Sequence classification

We evaluated DISTRO for English and Basque POS tagging using first-order sequence models.

English POS tagging has been long studied using the Penn Treebank data [15]. We used the standard train-test split [8, 24] – we trained on sections 0-18 of the Treebank and report performance on sections 22-24. The data is labeled with 45 POS labels. Some labels are semantically close to each other because they express variations of a base part-of-speech tag. For example, the labels NN, NNS, NNP and NNPS indicate singular and plural versions of common and proper nouns

We used the Basque data from the CoNLL 2007 shared task [17] for training the Basque POS tagger. This data comes from the 3LB Treebank. There are 64 fine grained parts of speech. Interestingly, the labels themselves have a structure. For example, the labels IZE and ADJ indicate a noun and an adjective respectively. However, Basque can take internal noun ellipsis inside noun-forms, which are represented with tags like IZE_IZEELI and ADJ_IZEELI to indicate nouns and adjectives with internal ellipses.

In both languages, many labels and transitions between labels are semantically close to each other. This observation has led, for example, to the development of the universal part-of-speech tag set

[19]. Clearly, the labels should not be treated as independent units of meaning and the model should be allowed to take advantage of the dependencies between labels.

| Language | Algorithm | Label Matrix Rank | Average accuracy (%) |
|---|---|---|---|
| English | Structured SVM | 45 | 96.2 |
| | DISTRO | 5 | 95.1 |
| | DISTRO | 20 | **96.7** |
| Basque | Structured SVM | 64 | 91.5 |
| | DISTRO | 58 | **92.4** |

Table 2: **Results on part-of-speech tagging.** The top part of the table shows results on English, where we see a 0.5% gain in accuracy. The bottom part shows Basque results where we see a nearly 1% improvement.

For both languages, we extracted the following emission features: indicators for the words, their prefixes and suffixes of length 3, the previous and next words and the word shape according to the Stanford NLP pipeline[2,3]. Table 2 presents the results for the two languages. We evaluate using the average accuracy over all tags. In the English case, we found that the performance plateaued for any label matrix with rank greater than 20 and we see an improvement of 0.5% accuracy. For Basque, we see an improvement of 0.9% over the baseline.

Note that unlike the atomic case, the learning objective for the first order Markov model is not even bilinear in the weights and the label vectors. However, in practice, we found that this did not cause any problems. In all but one run, the test performance remained consistently higher than the baseline. Moreover, the outlier converged to a much higher objective value; it could easily be identified. As an analysis experiment, we initialized the model with one-hot vectors (i.e. the baseline) and found that this gives us similar improvements as reported in the table.

# 6 Conclusion

We have presented a new model for structured output prediction called Distributed Structured Output (DISTRO). Our model is motivated by two observations. First, distributed representations for inputs have led to performance gains by uncovering shared characteristics across inputs. Second, often, structures are composed of semantically rich labels and sub-structures. Just like inputs, similarities between components of structures can be exploited for better performance. To take advantage of similarities among structures, we have proposed to represent labels by real-valued vectors and model compositionality using tensor products between the label vectors. This not only lets semantically similar labels share parameters, but also allows construction of complex structured output that can take advantage of similarities across its component parts.

We have defined the objective function for learning with DISTRO and presented a learning algorithm that jointly learns the label vectors along with the weights using alternating minimization. We presented an evaluation of our approach for two tasks – document classification, which is an instance of multiclass classification, and part-of-speech tagging for English and Basque, modeled as first-order sequence models. Our experiments show that allowing the labels to be represented by real-valued vectors improves performance over the corresponding structural SVM baselines.

**Acknowledgments**

We thank the anonymous reviewers for their valuable comments. Stanford University gratefully acknowledges the support of the Defense Advanced Research Projects Agency (DARPA) Deep Exploration and Filtering of Text (DEFT) Program under Air Force Research Laboratory (AFRL) contract no. FA8750-13-2-0040. Any opinions, findings, and conclusion or recommendations expressed in this material are those of the authors and do not necessarily reflect the view of the DARPA, AFRL, or the US government.

## Footnotes

[1]In a linear sequence model defined as a CRF or a structural SVM, these transition input features can simply be an indicator that selects a specific portion of the weight vector.

[2] http://nlp.stanford.edu/software/corenlp.shtml

[3] Note that our POS systems are not state-of-the-art implementations, which typically use second order Markov models with additional features and specialized handling of unknown words. However, surprisingly, for Basque, even the baseline gives better accuracy than the second order TnT tagger[5, 19].

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
