[Supplementary Material]

# Supplementary Material: Learning Distributed Representations for Structured Output Prediction

**Vivek Srikumar**
University of Utah
svivek@cs.utah.edu

**Christopher D. Manning**
Stanford University
manning@cs.stanford.edu

## Details of Alternating Minimization

This supplement gives details about the alternating minimization algorithm employed for training. Recall that the objective function is

$$f(\mathbf{w}, \mathbf{A}) = \frac{\lambda_1}{2}\mathbf{w}^T\mathbf{w} + \lambda_2||\mathbf{A}||_* + \frac{1}{N}\sum_i L(\mathbf{x}^i, \mathbf{y}^i; \mathbf{w}, \mathbf{A}) \tag{1}$$

In this paper, we instantiate $L$ to be the hinge loss, defined as

$$L(\mathbf{x}^i, \mathbf{y}^i; \mathbf{w}, \mathbf{A}) = \max_{\mathbf{y}} \left(\mathbf{w}^T\Phi_{\mathbf{A}}(\mathbf{x}^i, \mathbf{y}) + \Delta(\mathbf{y}, \mathbf{y}^i) - \mathbf{w}^T\Phi_{\mathbf{A}}(\mathbf{x}^i, \mathbf{y}^i)\right) \tag{2}$$

### Solving for the weight vector w

In steps 2 and 5 of the algorithm, when $\mathbf{A}$ is fixed to $\mathbf{A}^t$, the minimization of the function $f(\mathbf{w}, \mathbf{A}^t)$ is the same training a structural SVM. The objective function for these steps can be written as

$$\min_{\mathbf{w}} \frac{\lambda_1}{2}\mathbf{w}^T\mathbf{w} + \frac{1}{N}\sum_i L(\mathbf{x}^i, \mathbf{y}^i; \mathbf{w}, \mathbf{A}^t)$$

We use the stochastic sub-gradient descent algorithm for this minimization. Given a single example $\mathbf{x}^i$ labeled with the structure $\mathbf{y}^i$, the sub-gradient can be computed by performing standard loss-augmented inference as follows: $\hat{\mathbf{y}} = \max_{\mathbf{y}} \mathbf{w}^T\Phi_{\mathbf{A}}(\mathbf{x}^i, \mathbf{y}) + \Delta(\mathbf{y}, \mathbf{y}^i)$. The sub-gradient for this the objective defined over the single example with respect $\mathbf{w}$ is $\lambda\mathbf{w} + \Phi_{\mathbf{A}}(\mathbf{x}^i, \hat{\mathbf{y}}) - \Phi_{\mathbf{A}}(\mathbf{x}^i, \mathbf{y}^i)$. Note that gradient computation does not change from standard structural SVM because the feature representations are fixed. Loss-augmented inference also does not change because both $\mathbf{w}$ and $\mathbf{A}$ are fixed at inference time.

### Solving for the label matrix A

In Step 4 of the algorithm, we minimize f with respect $\mathbf{A}$ when $\mathbf{w}$ is fixed to $\mathbf{w}^{t-1}$. The objective function for this minimization is

$$f'(\mathbf{A}) = \lambda_2||\mathbf{A}||_* + \frac{1}{N}\sum_i L(\mathbf{x}^i, \mathbf{y}^i; \mathbf{w}^{t-1}, \mathbf{A}) \tag{3}$$

The objective $f'(\mathbf{A})$ can be written as $g(\mathbf{A}) + \lambda_2||A||_*$. The algorithm proceeds by taking a stochastic gradient step for $g$ followed by a proximal mapping to account for the regularizer, which we write as $h(\mathbf{A}) = ||A||_*$. The update has the following form:

$$A \leftarrow \mathbf{prox}_{th}\left(A - t\nabla g(A)\right)$$

Here, $\mathbf{prox}_{th}$ is the proximal operator, defined as

$$\mathbf{prox}_{th}(X) = \arg\min_U \left(||U||_* + \frac{1}{2t}||U - X||_F^2\right).$$

We will first look at the proximal operator before addressing the gradient computation.

We refer the reader to the monograph on proximal algorithms [1] for details about the method. In our case, the proximal operator with respect to the nuclear norm can be computed by performing singular value decomposition of the matrix $X = U\Sigma V^T$, where $\Sigma$ is the diagonal matrix of singular values $diag(\sigma_1, \sigma_2, \cdots)$. We have $\mathbf{prox}_{th}(X) = U\hat{\Sigma}V^T$. Here $\hat{\Sigma}$ is obtained by thresholding the singular values as follows:

$$\hat{\sigma_i} = \begin{cases} \sigma_i - t & \text{when } \sigma_i \geq t, \\ 0 & \text{when } |\sigma_i| \leq t, \\ \sigma_i + t & \text{when } \sigma_i \leq -t. \end{cases}$$

Even though the proximal mapping requires an SVD in the innermost loop, since the matrix is very small (of the order of the number of labels), this does not influence the training time adversely.

For the gradient step, we need to compute the gradient of $L(\mathbf{x}^i, \mathbf{y}^i; \mathbf{w}^{t-1}, \mathbf{A})$ with respect to $\mathbf{A}$. Since we are using the structured hinge loss, we use sub-gradient descent. As for the $\mathbf{w}$ case, we need to solve loss-augmented inference to get $\hat{\mathbf{y}}$. This leaves us with the problem of computing the gradient of $\mathbf{w}^T \Phi_{\mathbf{A}}(\mathbf{x}^i, \hat{\mathbf{y}}) - \mathbf{w}^T \Phi_{\mathbf{A}}(\mathbf{x}^i, \mathbf{y}^i)$ with respect to $\mathbf{A}$. We can compute the gradient symbolically by unrolling the recursive definition of the feature tensor function from Section 3.1.

Effectively, we need to compute the gradient of functions such as $\mathbf{w}^T \Phi_{\mathbf{A}}(\mathbf{x}, \mathbf{y})$ with respect to $\mathbf{A}$. From Equation (4), we have

$$\mathbf{w}^T \Phi_{\mathbf{A}}(\mathbf{x}, \mathbf{y}) = \sum_{p \in \Gamma_{\mathbf{x}}} \mathbf{w}^T vec\left(\Psi_p\left(\mathbf{x}, \mathbf{y}_p, \mathbf{A}\right)\right)$$

The term in the summation can be expanded as:

$$\mathbf{w}^T vec\left(\Psi_p\left(\mathbf{x}, \mathbf{y}_p, \mathbf{A}\right)\right) = \begin{cases} \mathbf{w}^T vec\left(\mathbf{a}_{l_y} \phi^T\right), & p \text{ is atomic,} \\ \mathbf{w}^T vec\left(\mathbf{a}_{l_{y_p}} \otimes \Psi_p\left(\mathbf{x}, \mathbf{y}_p^{1:}, \mathbf{A}\right)\right), & p \text{ is compositional.} \end{cases}$$
$$= \begin{cases} \sum_i \sum_j w_{dj+i} a_{l_y, i} \phi_j, & p \text{ is atomic,} \\ \sum_i \sum_j w_{dj+i} a_{l_{y_p}, i} \left[vec\left(\Psi_p\left(\mathbf{x}, \mathbf{y}_p^{1:}, \mathbf{A}\right)\right)\right]_j, & p \text{ is compositional.} \end{cases}$$

The final expression is simply the multi-linear expansion of the score assigned to a part $p$. If the part is atomic, the score is identical to the case discussed in Section 3.1 of the paper. For the compositional case, we use the property that if $a$ is a vector and $b$ is a tensor, we have $vec(a \otimes b) = vec(a \otimes vec(b))$. Our goal is to compute the gradient of the above expression with respect to each element of $\mathbf{A}$. For an atomic part, the gradient is simple to calculate. For a compositional part, the gradient can be computed using the chain rule. To complete the update for the label vectors, after the proximal step, we project each column of $A$ to the unit ball.

## References

[1] N. Parikh and S. Boyd. Proximal algorithms. *Foundations and Trends in optimization*, 1(3), 2013.