[Reviews · NeurIPS 2014]

Submitted by Assigned_Reviewer_3

The paper investigates finding an encoding of labels as real valued/low dimensional vectors (as opposed to indicator vectors) in Structured Output (SO) prediction setting. Tensor products are used for non-unary potentials. The label representations are learned jointly with weight vectors using an alternating optimization procedure, minimizing the trace norm of the label representation matrix and minimizing the hinge loss on the data wrt feature weights. Experiments on multiclass classification and sequence labeling are provided.

Efficient encoding of labels is a well studied problem, eg. in the context of multi-class, multi-label, multi-task learning. Within SO prediction framework, this has been explored in hierarchical classification, where the (not low dimensional) encoding of the labels are extracted from a given taxonomy. The paper nicely generalizes this idea when the predicted object can have arbitrary dependences using tensor products and adapts existing methods (alternating optimization procedure for low rank label matrix and feature weights). According to my knowledge, this had not been done for structured objects. The paper is clear and proposed method is sensible. While not very detailed, experiments show the validity of the approach. For the multiclass setting, comparison to a hierarchical SVM model [23] would be useful (while reminding the reader that such a taxonomy may not be available for many applications). It would also be interesting to show/mention the relations between inferred label representations (perhaps in an appendix).
Summary: Using techniques from literature on multilabel/multitask learning, generalizes SO prediction to low dimensional label encoding. While not on the high end of the originality spectrum, a useful and novel technique (to the best of my knowledge) that can be impactful researchers in the field.

Submitted by Assigned_Reviewer_10

Summary: The authors propose learning a distributed representation of outputs for structured prediction. Unlike multi-class embeddings, the label embeddings in this case need to be composable to form higher order factors. I.e. instead of a bilinear combination producing m*n values, we need a multi-linear combination producing m^2*n values for bigram factors, etc. The authors propose using the tensor product as a natural solution. Experimentally, they can learn POS tagging bigram models using low-rank tensors that outperform full rank explicit representations.

Major comments:

This paper is easy to understand at a high level, and the idea is very natural. In fact, using tensor decompositions for high order feature combinations is concurrently explored in the NLP community (see e.g. Low Rank Tensors for Scoring Dependency Structures, Lei et al. ACL2014). This is a nice explanation of how the idea can be applied to structured prediction in general. It will be immediately useful and interesting to anyone interested in structured prediction.

However, I felt the paper was very short on details. For instance, Algorithm 1 is so general as to be practically useless. It is literally just an algorithmic definition of alternating minimization. It would be much easier for other practioners to at least provide some equations showing that the gradient computations do not change; inference algorithms do not change, etc. Furthermore, there are a lot of practical concerns; how important is it to use a nuclear norm, instead of just assume fixed low rank and do standard SGD without needing proximal methods? What is impact on the training and evaluation time? In theory, low rank methods could be faster than traditional structured prediction, but the sparsity in 1-hot representations typically leads to very fast models, and this would sacrifice that speedup.

Also, more intuitive explanations for how this affects the complexity of the model would be helpful. For instance: how many parameters does one need to store for w? According to the description it seems like the size of w does not change; even if the features are shared between labels, the parameters are not explicitly shared. Is this correct? How does this method compare to applying low-rank constraints to w instead? Some discussion along these lines would be very helpful.

There are also other practical variations that would be interesting: e.g. learning parameters for both embedded features and non embedded features simultaneously, or allow for different embeddings for atomic vs. compositional labels.

Minor comments:

Since not everyone in the community is familiar with tensors, giving an explicit multi-linear form for a low-rank tensor parameterization (instead of just “vec …”) would be helpful.
Summary: This is a good paper, but it would have a much larger impact with more details and more intuitive discussion.

Submitted by Assigned_Reviewer_16

This is a very inspirational and well-written paper that challenges a fundamental assumption in many supervised classification tasks, namely that class labels are always distinct, discrete units of meaning.
Instead, the paper argues that class labels can expose common traits and that some labels are closer to each other than others, a fact that should be exploited during training.

Of course, this viewpoint is not entirely new -- such "distributional" arguments have been made in the connectionist literature before, regarding both input and output units, as correctly pointed out in the manuscript.

The main novelty in the present manuscript is then to extend such distributional formulations to structured prediction, where not only atomic labels but also compositional parts (such as labels of factors defined over multiple variables) can be modeled in a distributional manner.

Towards this end, a model is proposed in which labels of atomic parts are represented as vectors, and compositional parts as well as the corresponding feature spaces are defined in terms of tensor algebra. A traditional structured prediction model then corresponds to the special case where each atomic label is represented via one-hot encoding.

What makes the approach interesting is that both labels and model weights can be learned together in a joint objective function, based on the structured hinge loss. A rank-penalty regularization term is placed on the matrix of label vectors to encourage the labels to share weights. The parameters constituting the label vectors and the weights on features are trained by alternating minimization, which seems to work well most of the time, though the objective is not jointly convex.

The experiments demonstrate considerable improvements over a standard hinge loss formulation on both multiclass classification and sequence classification. Interestingly, the approach also allows to learn labels in a lower-dimensional vector space than the standard one-hot encoding, while maintaining excellent performance even for substantially reduced dimensionality.

Quality:
The paper is technically sound. The theory backing the approach is plausible and formally developed to a sufficient extent in the manuscript. The experiments are on the synthetic side, but still demonstrate the plausibility of the intuition underlying the paper convincingly.

Clarity:
The paper is well-written and the authors take care to give pedagogical examples to make the material easier to understand. A background in tensor algebra is still helpful when reading the paper.
Figure 1 runs over the right margin considerably and should be fixed.

Originality:
The extension of the distributional approach to compositional parts is novel to my knowledge.

Significance:
At present, I think neither the theory nor the practical tools (such as the learning algorithm, which seems to struggle with non-convexity now and then) are sufficiently developed for the approach to find broad adoption in the near future. Also, there needs to be further evidence that the proposed approach yields state-of-the-art results on established benchmark tasks. Nonetheless, the proposed approach is inspiring and seems to have a lot of potential.
Summary: This is a well-written and inspiring manuscript that follows a path that is different from, and orthogonal to, some of the techniques that enjoy particular popularity at the moment. Its potential is convincingly demonstrated, and as such, I think of it as a useful contribution that brings fresh ideas to the field of structured prediction and the technical program of NIPS.
Author Feedback
Author rebuttal: We thank the reviewers for their valuable feedback. We will add all
the suggested references, clarifications and corrections to the final
version.

Additionally, as per reviewer 1's suggestion, we will add a discussion
on practical concerns and the complexity of the model. In particular,
we will expand on the following brief responses to the questions
posed:

- We tried other regularizers (entrywise norms) and found that the
nuclear norm performs better, and has an intuitive explanation.

- The proximal method by itself does not add substantially to training
time because the matrices involved are small. Training time is,
however, affected by the density of the label vectors and the
reviewer is right in noting that the sparse 1-hot representation
leads to very fast training.

- The size of w depends on the dimensionality of the label vectors.
For eg, in the bottom part of table 1, the size of w is much smaller
than the top part because the size of the label vectors is fixed.

- Applying low-rank constraint on w was introduced by Srebro et al,
2005 and others for multiclass classification; with no obvious
natural extension for compositional parts. To account for this, our
model instead forces the label matrix to be low rank.

Reviewer 3 correctly points out the connection to
hierarchical/aspect-based encoding of multiclass classification. We
will add this.